# Cocktail Effect of Endocrine Disrupting Chemicals: Application to Chlorpyrifos in Lavender Essential Oils

**DOI:** 10.3390/ijerph191912984

**Published:** 2022-10-10

**Authors:** Sophie Fouyet, Elodie Olivier, Pascale Leproux, Sabrina Boutefnouchet, Mélody Dutot, Patrice Rat

**Affiliations:** 1Université Paris cité, CNRS CiTCoM, 75006 Paris, France; 2Laboratoires Léa Nature, 17180 Périgny, France; 3Yslab, Recherche & Développement, 29000 Quimper, France

**Keywords:** chlorpyrifos, lavender essential oil, P2X7 receptor, steroid hormones, polypeptide hormones

## Abstract

Chlorpyrifos is a pesticide that is toxic to human health and has been banned for the past decade. Due to its persistent and bioaccumulative properties, chlorpyrifos is still present in soil. Pregnant women can be exposed to chlorpyrifos through drinking water and herbal products, such as essential oils (EOs), resulting in adverse effects to the mother and fetus. Our objective was to evaluate and compare the potential endocrine disrupting effects of chlorpyrifos “free” or in contaminated lavender EO. We studied the release of four hormones and the activation of the P2X7 cell death receptor in human placental JEG-Tox cells as key biomarkers of endocrine toxicity for pregnant women (hPlacentox assay). We observed that “free” chlorpyrifos disrupted placental hormones and activated the P2X7 receptor, whereas chlorpyrifos in lavender EO disrupted only the placental hormones. We confirm that chlorpyrifos can be classified as an endocrine disrupting chemical (EDC) for pregnant women and point out that its endocrine disrupting effect may not be apparent when present in lavender EOs. Our results reveal the existence of specific reverse cocktail effects that may have protective properties against EDCs.

## 1. Introduction

Chlorpyrifos is one of the best known agrochemicals in the world. It is an organophosphate insecticide, miticide and acaricide used primarily to control foliage and soil insect pests, such as cockroaches and mosquitoes [1]. Although effective in controlling pests and thus improving crop yields, chlorpyrifos has been identified as toxic to human and animal health after acute and chronic exposures. Some studies indicate that chlorpyrifos, detected in urine and breast milk [2,3,4], can interfere with placental function and induce placental toxicity [5,6,7], resulting in reduced gestation length [8]. Chlorpyrifos toxicity has also been associated with endocrine disruption [9,10]. Due to the toxicity of chlorpyrifos, its use has been banned over the past decade in different parts of the world, including the United States and the European Union. However, due to its persistent and bioaccumulative properties, residues are still present in drinking water but also in organically grown products, such as fruits, vegetables, grains or essential oils (EOs) [11,12].

Lavender EO is one of the most popular EOs used in aromatherapy. Pregnant women use lavender EO to treat stress and anxiety, the most common psychiatric disorders during pregnancy, with prevalence rates of 11–17% [13,14]. In addition, pregnant women may be exposed to lavender EOs through a multitude of everyday products, including food flavorings, soaps, lotions, shampoos, hair products, colognes, laundry detergents and insect repellents [15].

First, we wondered whether chlorpyrifos is present in the lavender EOs to which pregnant women are exposed. Second, when pregnant women are regularly exposed to the endocrine disruptor chlorpyrifos through lavender EOs, is there a risk to them and their fetus? Considering the large proportion of pregnant women using potentially contaminated lavender EO, this would be a major public health problem. Not all pregnant women who use lavender EOs experience pregnancy problems, such as chlorpyrifos-induced shortening of pregnancy duration [8]. In this context, the question we raise is whether the endocrine effects of chlorpyrifos are the same when tested in a culture medium or in contaminated lavender EO. The objective of the present study was to evaluate the potential endocrine disrupting effects of chlorpyrifos in lavender EOs on the human placenta. Endocrine disrupting chemicals (EDCs) are defined by the World Health Organization (WHO) as exogenous substances or mixtures that alter the function(s) of the endocrine system and consequently cause adverse health effects (…) [16]. To achieve our objective, we used the hPlacentox test selected by the European public–private platform PEPPER for the pre-validation of endocrine disruptor characterization methods [17]. The hPlacentox assay recommends the use of the JEG-Tox human placental cell model to study hormonal disruptions (steroidal and polypeptide) and adverse health effects in these same cells [17,18] and thus meets the WHO definition of EDCs. The development of the hPlacentox assay was based in part on our previous work, showing that activation of the P2X7 receptor would be a common cellular mechanism of toxicity for EDCs in the placenta [19] and that there is a causal link between P2X7 receptor activation and EDC-induced hormone disruption.

## 2. Materials and Methods

### 2.1. Chemicals and Reagents

Minimum essential medium (MEM), fetal bovine serum (FBS), 2 mM glutamine, 10,000 U/mL penicillin and 10,000 µg/mL streptomycin, 0.05% trypsin-EDTA, and phosphate-buffered saline (PBS) were provided by Gibco (Paisley, UK) and cell culture plastics, such as flasks and microplates, by Corning (Schiphol-Rijk, The Netherlands). YO-PRO-1^®^ was obtained from ThermoFisher Scientific (Waltham, MA, USA) and alamar blue probes from Alfa aesar (Haverhill, MA, USA). The h-hCG and hPL ELISA kits were purchased from MyBioSource (San Diego, CA, USA) and the estradiol and progesterone kits from Cisbio (Codolet, France). Lavender (*Lavandula angustifolia* Mill.) EOs were obtained from Léa nature Laboratories (Périgny, France). We selected two different lavender EOs. One of the two lavender EOs came from France (Drôme) and was named lavender 1; the other one came from Bulgaria and was named lavender 2. The chlorpyrifos was purchased from Sigma-Aldrich (Saint Quentin Fallavier, France). Positive controls (Triton X-100, 4-tert-amylphenol (AP), diethylstilbestrol (DES), bisphenol A (BPA)) were used to ensure that the cell model reacts correctly under our experimental conditions and were obtained from Sigma-Aldrich.

### 2.2. EOs Composition Analysis

GC-MS analyses were performed using GCMS-QP2010 Ultra (Shimadzu Co., Kyoto, Japan) equipped with an Agilent 5973N MS detector. GC-FID analysis was performed using the Agilent Technologies 7820A GC-FID equipped with a Parker-Balston hydrogen generator. The analysis was developed using DB-5 (60 m × 0.25 mm × 0.25 µm) and DB-WAX (60 m × 0.25 mm × 0.25 µm) columns for GC-MS analysis and DB-WAX (60 m × 0.25 mm × 0.25 µm) for GC-FID analysis.

Amounts of 1% of solutions of lavender 1 and lavender 2 EOs in ethyl acetate were prepared and injected for analysis. The following analytical conditions were used for the instruments and columns: the oven temperature was programed at 50 °C, held for 5 min and then increased to 240 °C at a rate of 3 °C/min; the injector temperature was set at 240 °C; the carrier gas was helium with a flow rate of 1 mL/min; a fractionation ratio of 1:50 was applied; and the injection volume was set at 1 μL.

For the GC-MS mass spectrometry interface, the MS source temperature was set at 220 °C with an ionization energy of 70 eV and an interface temperature of 240 °C. A full scan was recorded (50–700 m/z).

Constituent identification was performed by comparing the GC-MS mass spectra with the components in the mass spectra library (NIST 98) and comparing the retention indices (RI) calculated from the injection of the C8–C20 alkane hydrocarbon mixture with the literature retention indices on the DB-5 column using the following formula [20]:

RI = (100 × n) + 200 × ((RTi − RTn-1)/(RTn − RTn-1)) where the retention indices RI are calculated using the retention time of each compound (RTi) and the retention time of the C8 to C20 alkanes, preceding (RTn-1) and following (RTn) the considered peak.

### 2.3. EOs Pesticide Analysis

The analysis of pesticides was subcontracted to the service provider Phytocontrol. In total, 250 pesticides were analyzed by GS-MS-MS, with a limit of quantification of 10 ppb.

### 2.4. Human Placental Cell Culture

The JEG-3 human trophoblast cell line was obtained from the American Type Culture Collection (ATCC HTB-36). Cells were cultured in minimal essential medium (MEM) supplemented with 10% fetal bovine serum (FBS), 1% L-glutamine, 0.5% penicillin and streptomycin in 75 cm^2^ polystyrene flasks. Cell cultures were maintained in a cell culture incubator (37 °C, saturated humidity, 5% CO_2_). When JEG-3 cells reached subconfluence, they were detached with trypsin-EDTA and counted. The cell suspension was diluted and seeded into 96-well microplates at a cell density of 80,000 cells/mL (200 µL/well) and stored at 37 °C for 24 h.

### 2.5. Cell Incubation

Stock solutions of lavender EOs were obtained after 2/3 dilution in ethanol and then diluted in MEM supplemented with 2.5% FBS to obtain three concentrations: 0.17 × 10^−3^%, 0.17 × 10^−2^% and 0.17 × 10^−1^% (*v*/*v*). Cells were incubated for 72 h with the different concentrations of EOs, according to the literature [21,22], in MEM supplemented with 2.5% FBS according to the protocol of Olivier et al. [18]. Based on our expertise, we previously suggested that the serum concentration in JEG-3 cells be reduced to 2.5% FBS for toxicology studies. JEG-3 cells prepared in reduced FBS were renamed JEG-Tox cells. Chlorpyrifos was diluted under the same conditions as lavender EOs and tested at similar percentages as found in EOs, corresponding to 0.23 × 10^−7^ mg/mL, 0.23 × 10^−6^ mg/mL and 0.23 × 10^−5^ mg/mL. The final concentration of ethanol in the cells was less than or equal to 0.008%.

### 2.6. Cell Viability. Alamar Blue Assay

The alamar blue stock solution (0.1 mg/mL) was prepared in PBS buffer and stored at 4 °C, protected from light. The working solution used for the assay was obtained by diluting the stock solution 1/11 in a culture medium supplemented with 2.5% FBS. After removing the supernatants and rinsing the cells with PBS, the amber blue working solution was dispensed into the wells. Then, the microplate was placed in the incubator for 6 h and read (λex = 535 nm and λem = 600 nm) with the Tecan Spark^®^ microplate reader (Männedorf, Switzerland). Triton^®^ X-100 was used as a positive control for cytotoxicity in the alamar blue assay.

### 2.7. Steroid and Polypeptide Hormones Quantification

After centrifugation of the 96-well microplates, cell supernatants were collected, and hormones were quantified—h-hCG and hPL by ELISA, according to the supplier’s instructions (MyBioSource, Vancouver, BC, Canada), and estradiol and progesterone by FRET, according to the supplier’s instructions (Cisbio, Codolet, France). The Spark^®^ microplate reader was used for both techniques. Substances of very high concern (SVHC) with endocrine disrupting properties were used as positive controls: BPA for estradiol release, AP for progesterone release and DES for h-hCG and hPL release.

### 2.8. Cell Death P2X7 Receptor Activation: YO-PRO-1^®^ Assay

Activation of the P2X7 cell death receptor was assessed using the YO-PRO-1^®^ assay [23]. The YO-PRO-1^®^ probe enters cells only after the pore opening induced by P2X7 receptor activation and binds to DNA by emitting fluorescence. A stock solution of 1 mM YO-PRO-1 was diluted 1:500 in PBS just prior to use and dispensed into the microplate wells. After an incubation time of 10 min at room temperature, the fluorescence signal was read (λex = 485 nm, λem = 531 nm) using the Spark^®^ microplate reader. BPA was used as a positive control for P2X7 receptor activation in JEG-Tox cells [19,24].

### 2.9. Results Exploitation and Statistical Analysis

Results are expressed as percentage or fold change relative to control cells and presented as means of at least three independent experiments ± standard errors of the mean. The normal distribution of the data was confirmed by the D’Agostino–Pearson test. Statistical analysis was performed using GraphPad Prism 8 software (San Diego, CA, USA). One-way analysis of variance followed by Dunnett’s test with risk α set at 5% was performed to compare chlorpyrifos and lavender EOs containing chlorpyrifos and incubation with the negative control (*p*-values expressed as *). The significant levels were * *p* < 0.1, ** *p* < 0.01, *** *p* < 0.001.

## 3. Results

### 3.1. Analysis of Lavender EOs Composition

We analyzed the composition of two lavender EOs (Table 1). After analysis of 250 pesticides (Appendix A), we could observe that these two samples were contaminated by chlorpyrifos at 0.014 mg/mL.

### 3.2. Cell Viability

We investigated the viability of JEG-Tox cells, using the alamar blue assay, after incubation with chlorpyrifos and the two lavender EOs containing chlorpyrifos.

The two lavender EOs (1 and 2) were tested at three concentrations: 0.17 × 10^−3^%, 0.17 × 10^−2^% and 0.17 × 10^−1^%. These concentrations were chosen according to the literature [21,22]. Chlorpyrifos concentrations were chosen based on the amounts present in the EOs tested, 0.23 × 10^−7^ mg/mL, 0.23 × 10^−6^ mg/mL and 0.23 × 10^−5^ mg/mL, since the two lavender EOs tested contained 0.014 mg/kg chlorpyrifos. Any concentration that induced a loss of cell viability greater than or equal to 30% was considered cytotoxic [25] and discarded for further analysis involving hormones and the P2X7 receptor.

**Table 1 ijerph-19-12984-t001:** Composition and pollutants analysis of lavender essential oils (EOs) (mass spectrometry (MS), retention indices (RI), retention time (RT; min), standard (st)).

Components					Lavenders Tested		
	RT DB-Wax	RT DB-5	RI DB-5 Calc	RI DB-5 Litt [26]	Lavender 1 %	Lavender 2 %	
α-pinene	12.190	5.444	874.818	940–945	0.284	0.356	MS, RI
β-pinene	16.067	7.929	977.653	975–985	1.044	1.127	MS, RI
d-limonene	21.371	9.52	1023.578	1030–1035	0.482	0.460	MS, RI, st
cryptone	22.010	16.930	1176.547	1160–1200	-	0.280	MS, RI
1.8-cineole	22.282	9.679	1025.248	1010–1045	0.427	0.513	MS, RI
cis-β-ocimene	23.700	10.112	1034.283	1031–1041	2.848	5.072	MS, RI
trans-β-ocimene	24.997	10.585	1044.152	1045–1060	3.066	3.39	MS, RI
3-octanone	25.190	7.743	969.956	965–990	1.665	1.402	MS, RI
hexyl acetate	26.715	9.056	1012.248	1000–1022	0.425	0.613	MS, RI
1-octen-1-ol acetate	36.080	13.701	1109.171	-	1.208	1.291	MS, RI
3-octanol	37.788	8.306	993.254	990–1000	0.218	0.201	MS, RI
hexyl butyrate	39.888	17.653	1191.633	1190–1195	0.347	0.343	MS, RI
1-octen-3-ol	43.410	7.480	961.845	960–980	0.388	0.251	MS, RI
camphor	49.980	15.013	1136.547	1120–1140	0.172	-	MS, RI, st
linalool	54.391	13.290	1100.595	1080–1110	34.615	35.813	MS, RI, st
linalyl acetate	54.540	20.400	1252.11	1240–1260	34.333	28.747	MS, RI, st
α-santalene	56.586	27.440	1409.507	1410–1425	0.417	0.364	MS, RI
β-caryophyllene	58.677	27.272	1405.326	1410–1420	3.339	3.58	MS, RI
terpinen-4-ol	59.277	16.746	1172.707	1160–1180	3.057	4.788	MS, RI, st
lavandulyl acetate	59.550	21.914	1285.74	1270–1290	3.488	3.639	MS, RI, st
nd	63.320	28.743	1425.485	-	1928	3.044	MS, RI
Lavandulol	63.894	16.190	1161.106	1155–1175	1.07	1.13	MS, RI
α-terpineol	65.564	17.474	1187.898	1180–1195	1.144	1.119	MS, RI, st
Borneol	65.871	16.289	1163.172	1162–1175	0.784	0.356	MS, RI
germacrene D	66.035	29.829	1468.965	1460–1495	0.385	-	MS, RI
geranyl acetate	66.968	26.035	1377.277	1370–1390	0.816	0.737	MS, RI
Geraniol	72.310	18.996	1220.924	1260–1265	0.044	0.138	MS, RI
caryophyllene oxide	77.345	33.695	1565.182	1580–1585	0.129	-	MS, RI
TOTAL					97.079	98.398	
Pesticides					mg/mL		
Chlorpyrifos					0.014	0.014	

Neither chlorpyrifos nor lavender EOs containing chlorpyrifos had cytotoxic effects at the concentrations tested (Figure 1).

### 3.3. Hormone Release

Lavender 1 reduced progesterone secretion (×0.82 to 0.17 × 10^−^^3^% and ×0.83 to 0.17 × 10^−^^1^% vs. control), whereas lavender 2 and chlorpyrifos significantly increased progesterone secretion to 0.17 × 10^−^^1^% (×1.31 vs. control, Figure 2a) and 0.23 × 10^−^^6^ mg/mL (×1.51 vs. control), respectively.

Lavender 1 induced estradiol release (×1.39 to 0.17 × 10^−^^2^% and ×1.32 to 0.17 × 10^−^^1^%, Figure 2b). Chlorpyrifos significantly reduced estradiol release (×0.77 to 0.23 × 10^−7^ mg/mL and ×0.70 to 0.23 × 10^−^^5^ mg/mL), whereas lavender 2 had no effect on estradiol release.

Chlorpyrifos had no effect on h-hCG release, whereas lavender 1 and lavender 2 disrupted it; lavender 1 induced a significant increase in h-hCG release (×1.36 vs. control at 0.17 × 10^−^^1^%), and lavender 2 reduced h-hCG secretion (×0.77 to 0.17 × 10^−1^%, Figure 2c).

Lavender 1 had no effect on hPL release. Lavender 2 slightly upregulated hPL (×1.12 to 0.17 × 10^−^^3^% and ×1.20 to 0.17 × 10^−^^2^%), while chlorpyrifos induced a significant reduction in hPL release compared to the control at 0.23 × 10^−^^5^ mg/mL (×0.64, Figure 2d).

### 3.4. P2X7 Receptor Activation

P2X7 pore opening, reflecting P2X7 receptor activation, was assessed using the fluorescent YO-PRO-1^®^ assay. Two lavender EOs had no effect on P2X7 receptor activation, while chlorpyrifos statistically activated the P2X7 receptor at 0.23 × 10^−^^6^ mg/mL and 0.23 × 10^−5^ mg/mL (×1.17 and ×1.16, respectively, Figure 3).

## 4. Discussion

The objective of the present study was to explore the potential endocrine disrupting effect of chlorpyrifos, “free” or incorporated in lavender EOs, on human placental cells.

According to the WHO definition, EDCs are exogenous substances or mixtures that alter the function(s) of the endocrine system and consequently cause adverse health effects in an intact organism, or its progeny, or (sub)populations [16]. We adapted this definition to in vitro models and chose to study P2X7 receptor activation in JEG-Tox placental cells because this receptor is known to trigger placental disorders, such as preterm birth and preeclampsia [27,28], with dramatic consequences for the mother [29] and the unborn child [30]. Considering P2X7 receptor activation as the trigger for adverse health effects, we chose the hPlacentox assay to study the release of progesterone, estradiol, h-hCG and hPL in cell supernatants and P2X7 receptor activation in the JEG-Tox cell model of human endocrine placenta. Cells were incubated with chlorpyrifos at the same concentrations as those present in two different lavender EOs.

The present study demonstrated that chlorpyrifos does not behave in the same manner, “free” or incorporated into lavender EOs, in hydrophilic cell culture medium in human placental JEG-Tox cells.

First, we observed that chlorpyrifos and the two studied lavender EOs containing chlorpyrifos disturbed at least one of the placental hormones. Chlorpyrifos significantly stimulated progesterone release and reduced estradiol and hPL release. Lavender 1 triggered a large release of estradiol and h-hCG. Lavender 2 increased progesterone and hPL, and reduced h-hCG secretion. In addition, two lavender EOs without chlorpyrifos (3 and 4) were included in our study. Lavender 3 had no effect on the four hormones tested, whereas lavender 4 reduced estradiol release (Appendix A). The variable and complex compositions of lavender EOs could lead to variable effects in the placenta and explain the different hormonal perturbations observed. We could notice that the surface ratio of linalool and linalyl acetate, the major components, was slightly different between the two lavender EOs, with linalyl acetate being present in a lower proportion in lavender sample 2. In addition, camphor, germacrene D and caryophyllene oxide were not detected in lavender 2, while cryptone was only identified in lavender 1. The chromatographic profile of EOs can be different depending on many environmental factors, such as climate, weather conditions and altitude [31,32]. Lavender 1 and 3, from the Drôme region of France, grew at 600 to 1200 m above sea level, whereas lavender 2 and 4, from Bulgaria, grew at an average altitude of 470 m.

Second, we observed that chlorpyrifos induced significant activation of the P2X7 receptor, unlike the four lavender EOs tested, even the two that were contaminated with chlorpyrifos, at each concentration tested (Figure 3). Interestingly, lavender EOs, whether contaminated with chlorpyrifos (lavender 1 and 2) or not (lavender 3 and 4, Appendix A), shared the same lack of effects on P2X7 receptor activation.

Based on these results, several conclusions can be drawn. First, the incubation of JEG-Tox cells with chlorpyrifos resulted in both placental hormone disruption and P2X7 receptor activation. Chlorpyrifos meets the WHO definition of EDCs and can be considered an EDC, causing hormone disruption and deleterious health effects. These results are in agreement with the literature [9,10]. Second, the endocrine disrupting effects of chlorpyrifos were not revealed in lavender EOs. The observed differences may be due to interactions between chlorpyrifos and EO components. The cocktail effects and synergistic interactions between chemicals in mixtures are well documented in the literature. Many chemicals can potentially be more toxic if mixed [33]; this is the case for EDCs [34]. In our study, we observed a reverse cocktail effect, i.e., instead of being more toxic in a mixture, chlorpyrifos appeared less toxic in lavender EOs in terms of the endocrine disruption effects. The different components present in EOs could act as vectors capable of modifying the pharmacological and toxicological mechanisms induced by chlorpyrifos, for example, by decreasing its bioavailability or by reducing access to its biological target, thus reducing the toxic effects of chlorpyrifos. For example, linalool has low solubility, resulting in low bioavailability [35]. Linalool could interact with chlorpyrifos and decrease its own bioavailability. Lavender EOs could therefore have a protective effect against the pesticide they might contain. Further studies are needed to understand the interactions between the molecules composing the EOs, the pollutants they contain and the observed “protective” effects that we call here the reverse cocktail effect. Moreover, it would be very important to verify whether the reverse effect is also present in lavender EOs of other origins, as well as in other EOs.

Our study confirms the difficulty and complexity involved in drawing conclusions on the endocrine disruption potential of a molecule. The endocrine disrupting effects of chemicals can vary depending on their arrangement and embedding. Well-described synergistic cocktail effects increase the toxicity of chemicals, including EDCs. Here, we highlight the potential existence of an inverse cocktail effect with chlorpyrifos in lavender EOs, which may have a potential protective effect for pregnant women. As this study was conducted on four samples of lavender EOs, further studies are needed to draw a full conclusion on these first results.

## 5. Conclusions

In conclusion, this study confirms that chlorpyrifos can be classified as an EDC for pregnant women, and we point out that its endocrine effect may not appear when present in lavender EO. Our results reveal the potential existence of an unexpected reversible cocktail effect, which may have protective properties against EDCs.

## Figures and Tables

**Figure 1 ijerph-19-12984-f001:**
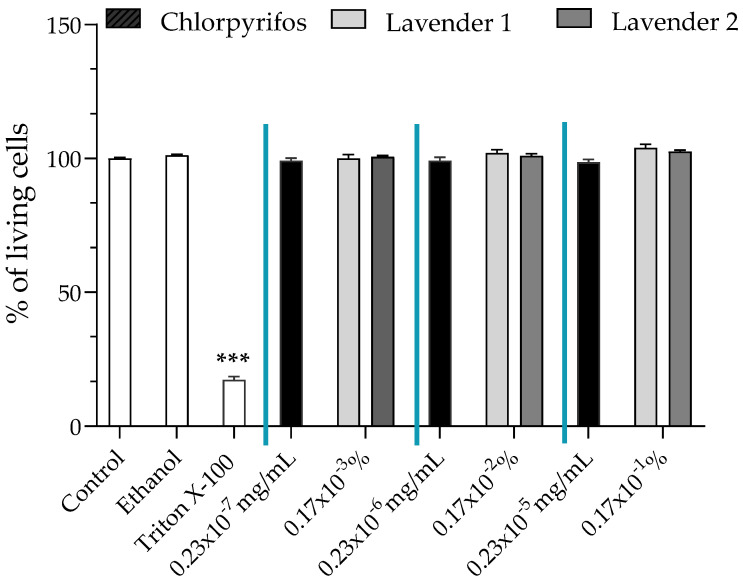
Cell viability was evaluated using the alamar blue assay after incubation of JEG-Tox cells with chlorpyrifos at 0.23 × 10^−^^7^ mg/mL, 0.23 × 10^−^^6^ mg/mL and 0.23 × 10^−^^5^ mg/mL or two lavender essential oils (EOs 1 and 2) at 0.17 × 10^−^^3^%, 0.17 × 10^−^^2^% and 0.17 × 10^−^^1^% for 72 h. Triton^®^ X-100 at 0.016% was used as a positive control. Significance levels were *** *p* < 0.001.

**Figure 2 ijerph-19-12984-f002:**
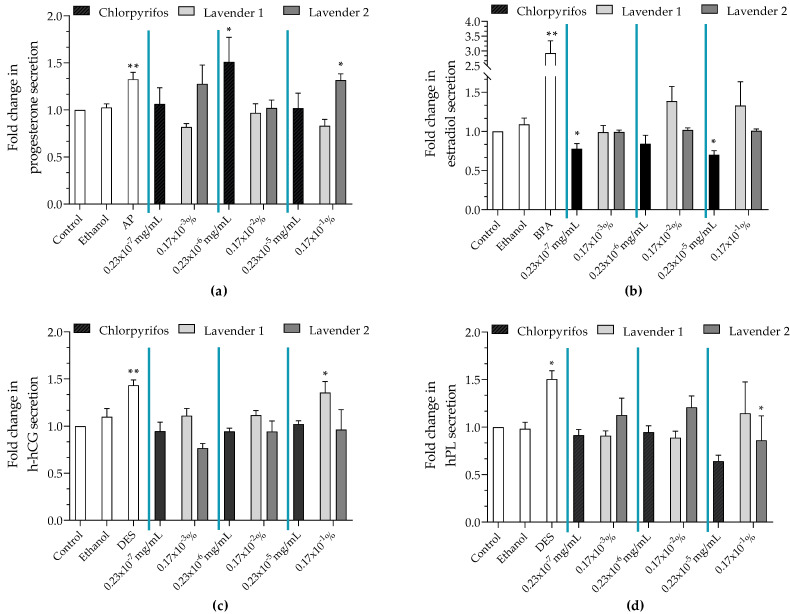
Quantification of the release of (**a**) progesterone, (**b**) estradiol, (**c**) h-hCG and (**d**) hPL after incubation of JEG-Tox cells with chlorpyrifos 0.23 × 10^−^^7^ mg/mL, 0.23 × 10^−^^6^ mg/mL and 0.23 × 10^−^^5^ mg/mL or two lavender EOs (1 and 2) at 0.17 × 10^−^^3^%, 0.17 × 10^−^^2^% and 0.17 × 10^−^^1^% for 72 h. AP at 10 µM, BPA at 20 µM, DES at 3.75 µM was used as a positive control. The significance levels were * *p* < 0.1 and ** *p* < 0.01.

**Figure 3 ijerph-19-12984-f003:**
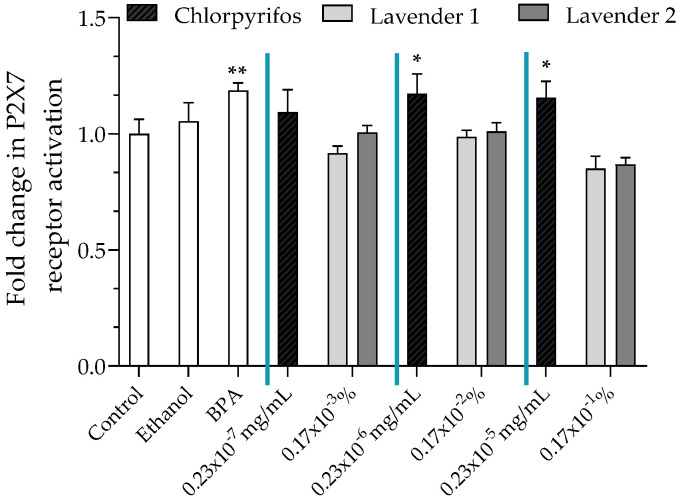
P2X7 receptor activation was evaluated using the YO-PRO-1^®^ assay after incubation of JEG-Tox cells with chlorpyrifos at 0.23 × 10^−^^7^ mg/mL, 0.23 × 10^−^^6^ mg/mL and 0.23 × 10^−^^5^ mg/mL or two lavender EOs (1 and 2) at 0.17 × 10^−^^3^%, 0.17 × 10^−^^2^% and 0.17 × 10^−^^1^% for 72 h. BPA at 20 µM was used as a positive control. The significance levels were * *p* < 0.1 and ** *p* < 0.01.

## Data Availability

The data presented in this study are available on request from the corresponding author.

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
