# Peer review of "Cocktail Effect of Endocrine Disrupting Chemicals: Application to Chlorpyrifos in Lavender Essential Oils"

_ijerph, 2022, doi:10.3390/ijerph191912984_

Round 1
Reviewer 1 Report (Previous Reviewer 1)
Professional English editing/proofreading is needed to enhance the overall manuscript presentation.
Be consistent in using abbreviations throughout the text.
Line 72-76 is not recommended to put in Introduction.
Line 116: Cite a reference for the formula
GC or GC/MS is used for calculation of RIs? Please make sure the right instrument and column are used for the RIs calculation.
What is the purpose of using DB-Wax column? No RIs calculation was involved nor comparison between the columns used.
References for RI DB-5 Literature in Table 1?
Confusing terms, France 1 and Bulgaria 2 OR lavender 1 and lavender 2? Please be consistent.
Author Response
Please see the attachment

Reviewer 2 Report (Previous Reviewer 2)
Thank You for improving the manuscript as advised.
Author Response
The reviewer has no new comments.
This manuscript is a resubmission of an earlier submission. The following is a list of the peer review reports and author responses from that submission.
Round 1
Reviewer 1 Report
The manuscript “Towards a new concept: the reverse cocktail effect of lavender essential oils and endocrine disruptor chlorpyrifos in human placental cells” requires major improvement.
Authors should revise and improve the English language and grammar throughout the manuscript.
Please organise and update the introduction. Too many small paragraphs. Authors are recommended to consolidate the content and to include related recent articles.
Once abbreviations were introduced, please use them throughout the text. What is EDCs? Some abbreviations are not being introduce in full name. There are too many abbreviations, I’d recommend to include all the abbreviations in a table.
Line 42: Chlorpyrifos is detected in almost all women. Please explain why only in women? Any rationalisation or justification?
Line 65: (…) what does it mean?
Materials and methods are not organised. Authors should divide them into sub-sections (with title) consisting of different part of experimental. Chemicals and reagents names do not need to be in capital letters, ed. Fetal Bovine Serum. Authors are advised to revise and check thoroughly. In addition, chemicals and reagents’ manufacturer and country should be included.
Which column was used? DB-5 or DB-Wax? How many replicate was used for each lavender oil? What is the method used for the calculation of %constituent in the lavender oil? Concentration used in the GC and GC/MS analysis? Any dilution?
There is no experimental/method/parameter for EOs pesticide analysis.
Lack of critical information on the composition of lavender oils (Table 1). Please include the retention time, experimental and literature retention indices for each identified constituent. Please include the chromatogram in the supplementary materials.
Authors include α-and β-pinene as one constituent. Did they elute in the same peak? In fact, they have different RIs, should be easily differentiated.
Serious mistakes were found. The total % of Spontaneous lavenders is not 96.313%!
Section Discussion needs a major touch up. Statements were found repeating. Contents are found unorganised. No reference for Line 220-225. There is no need to repeat the objectives again the Discussion. Line 248-250, concentration shouldn’t be used to indicate the differences, as no quantification has been done in the present study. No comparison of EO composition with other lavender oils. Lack of sample size in the present study to make a comprehensive finding.
Line 290: Patents? It is empty.
Reference style is not consistent. Please revise.
